# Estimation of Lower Limb Kinematics during Squat Task in Different Loading Using sEMG Activity and Deep Recurrent Neural Networks

**DOI:** 10.3390/s21237773

**Published:** 2021-11-23

**Authors:** Alireza Rezaie Zangene, Ali Abbasi, Kianoush Nazarpour

**Affiliations:** 1Department of Biomechanics and Sports Injuries, Faculty of Physical Education and Sports Sciences, Kharazmi University, Tehran 15719-14911, Iran; Std_Alireza.Reyzan@khu.ac.ir; 2School of Informatics, The University of Edinburgh, Edinburgh EH8 9AB, UK; kianoush.nazarpour@ed.ac.uk

**Keywords:** surface electromyography (sEMG), continuous estimation, deep neural networks (DNNs), joint angle estimation, squat

## Abstract

The aim of the present study was to predict the kinematics of the knee and the ankle joints during a squat training task of different intensities. Lower limb surface electromyographic (sEMG) signals and the 3-D kinematics of lower extremity joints were recorded from 19 body builders during squat training at four loading conditions. A long-short term memory (LSTM) was used to estimate the kinematics of the knee and the ankle joints. The accuracy, in terms root-mean-square error (RMSE) metric, of the LSTM network for the knee and ankle joints were 6.774 ± 1.197 and 6.961 ± 1.200, respectively. The LSTM network with inputs processed by cross-correlation (CC) method showed 3.8% and 4.7% better performance in the knee and ankle joints, respectively, compared to when the CC method was not used. Our results showed that in the prediction, regardless of the intensity of movement and inter-subject variability, an off-the-shelf LSTM decoder outperforms conventional fully connected neural networks.

## 1. Introduction

Amputation, that is a removal of a limb, can be because of a trauma or medical illness. In the United States about 1.6 million people were amputated in 2005 and it is expected to grow to than 3.5 million by 2050 [1]. Amputees traditionally use passive prostheses to compensate for missing limb function. Most of lower limb amputees have asymmetrical gait as a nature of their prosthetic limb [2]. Asymmetrical gait leads to degenerative changes in intact limb joints [3] and lumber spine [4], osteoarthritis in the intact limb [4,5], osteoporosis in the prosthetics limb [4], pain in the residual limb [6], and low back pain [7,8]. The symmetrical function of the lower limbs depends on the kinematics similarity of the prosthesis to their biological counterparts. Therefore, the symmetry between the two limbs can be improved through the kinematic adjustments of the prosthesis. It is expected that active prostheses improve gait symmetry and walking stability [9], reduce metabolic cost [10], and enhance similarity of ground reaction forces to intact limb [11]. However, according to Winter and Sienko (1988), because there is no connection between a passive prosthesis and the central nervous system (CNS), even assuming symmetry in the kinematic motion pattern, asymmetry will appear elsewhere in the system [12]. Therefore, even in the active type of prosthesis, lack of connection between prosthesis and human nervous system, can lead to some degree of asymmetry in unilateral lower limb amputee and there must be some kind of interaction between prosthesis and the user.

The main purpose of brain-computer interface (BCI) is to provide control and communication channels that are independent of brain’s output channels of peripheral muscles and nerves [13]. Bioelectric signals such as surface electromyographic (sEMG) signals can be used to control bionic lower-limbs [13]. The sEMG captures from local muscles and it contains less noise compared to EEG signals. The sEMG signals record neuromuscular activity 20–200 ms before the start of specific movement [14] and it can be used to predict the body’s movement initiation [15]. Prostheses that are working based on sEMG signal are called myoelectric prostheses. Previous research has focused on controlling these prostheses in everyday movement such as level ground walking [16], ascending or descending stairs [17]. However, the use of these prostheses can be extended to more complex activities such as athletic movements.

Lower limb amputation has a negative effect on athletic performance in amputee due to asymmetry between the two limbs [18]. Previous body of literature suggests that the level of physical activity in the lower limb amputee is lower than healthy population [19,20,21]. Participation in athletic tasks has benefits in terms of increasing muscle force, body mass, improving the function of the cardiorespiratory system, and it has positive effect on self confidence in amputee population [22]. It can have a significant impact on the psychological consequences after amputation, including discomfort, anxiety, and fatigue [23]. Myoelectric prostheses can be used in complex and intense athletic movements through a control scheme that is suitable for sports activities. Prostheses that can be used in athletic conditions can reduce the limitations of amputation in sports activities and solve many of the mentioned problems. Artificial intelligence can be used to develop a control scheme that is suitable for this purpose.

Artificial neural networks are a group of machine learning algorithms that are widely used in classification and regression problems. In previous studies, these networks have been used to estimate kinematic and kinetic parameters through sEMG signal. The mapping that makes this prediction can be implemented in the control of myoelectric prosthesis [24], exoskeleton robots [25], and rehabilitation robots [26]. Different algorithms have used by researchers to estimate the joint angles by sEMG during different tasks. For instance, Farmer et al. (2014) used a nonlinear autoregressive exogenous model to estimate continuous ankle joint angle by selected muscle sEMG signals [27]. Chen et al. (2018) suggested the use of deep belief network for feature selection and back propagation algorithm to estimate lower limb joint angle during normal walking [28]. Wang et al. (2019) used Elman network in continuous estimation of knee joint angle during simple flexion/extension task and they used a new feature extraction method based on Wavelet Transform and Correlation Dimension called correlation dimension of wavelet coefficient [29]. Wang et al. (2020) presented an extreme learning machine to predict lower limb kinematics and their focus was on rehabilitation movements under normal condition [30]. Doğukan Keleş et al. (2020) investigated a time delay feed-forward multilayer network for ankle angle and moment estimation during level ground walking and they have considered different combinations of lower limb muscles with the aim of acquiring the best performance by the least number of muscles in estimating the target parameters [31]. In all of the researches reviewed above the classical networks have used for mapping input to target space. The sEMG signals and Kinematics data are time series that have sequential structures. Each data in these timeseries depends on previous data with different time step. Therefore, recurrent networks are good choice for implementation on biomechanical datasets. However, classical recurrent networks have two main drawbacks-short-term memory and explode or vanishing of gradients- that have been solved in deep network.

Deep learning algorithms, e.g., convolutional neural networks (CNNs), recurrent neural networks (RNNs) and generative adversarial networks (GANs), are commonly used to analyse signal and image data [32,33]. RNNs are classified into two subgroups: gated recurrent unit (GRU) [34] and long-short term memory (LSTM) network [35]. These algorithms are inherently designed for data that has a sequential structure. LSTM was introduced earlier than GRU but it has a more complex structure. Through the gates designed in the LSTM cell, two main problems of classical recurrent algorithms that we mentioned earlier have been solved. The use of deep networks for prediction limb movement compared to classical networks has been limited. In a study conducted by Xia et al. (2017) a combination of an RNN and a CNN was used for upper limb motion prediction. In comparison with support vector regression, their proposed algorithms achieved more accuracy and better results [36]. In a similar research, Chen et al. (2019) used an LSTM network for continuous estimation of upper limb angular kinematics. Their study showed LSTM has much better performance compared to multi-layer perceptron (MLP) which is also a shallow network (13.57% reduction in the RMSE metric) [37]. To the best of our knowledge, none of the previous studies have examined the use of deep networks for predicting lower limb kinematics using sEMG signal in athletic condition.

The sEMG signal is a non-stationarity signal which its frequency spectrum changes with fatigue [38]. Two approaches can be used to predict kinematic parameters in athletic conditions using this signal: (1) The use of classified fatigue or non-fatigue conditions and activity intensity and train a separate network for continuous estimation in each class. (2) Train a network that is robust to the intensity of performing movements and frequency spectrum changes due to fatigue and independently of these factors predicts the target parameter. The first approach is very complex and requires both pattern recognition and continuous estimation. In this paper, the second approach is considered. The purpose of this research is as follows:First, train a deep-RNN that can predict lower limb kinematics independent of intensity of the movement.Second, inter subject prediction for evaluation the generalizability of the network.Third, comparison of a deep-RNN performance with a common classical network in task performance.

## 2. Materials and Methods

### 2.1. Data Collection

In this case, 19 trained body builders with no history of lower extremity injuries volunteered to participate in this research (age 24.50 ± 5.34 year, mass 90.80 ± 13.45 kg, height 180.65 ± 5.48 cm). The main aim of this study was to predict joints kinematics from sEMG in real loading situation and trained body builder were chose because they were familiar with squat task in real different loading situation. They signed an informed consent form. The protocol was approved by the Kharazmi University Institutional Review Board.

Squat movements in four loading conditions were considered as the selected movement in this study due to its importance in many athletic activities. At the beginning of the test, five-repetition maximum (5RM)-the maximum weight that can be lifted five times-in squat movement was recorded for each participant. After 48 h and complete recovery, each participant was required to complete five squats at no loading condition, 60%, 80%, and 100% of their 5RM with 2 min rest between sets. During each trial, the sEMG signals of vastus medialis (VM), rectus femoris (RF), biceps femoris (BF), tibialis anterior (TA), and medial gastrocnemius (MG) muscles of right leg were recorded with a wireless sEMG collection device (myoMuscle, Noraxon USA Inc., Scottsdale, AZ, USA) at the sampling rate of 1500 Hz which they are the most muscles move the knee and ankle in the sagittal plane [39]. To avoid information redundancy, except for the knee extensor muscle group, we selected only one muscle from the other muscle groups involved in the movement. Since redundancy reduces prediction accuracy and eliminating of redundancy also increases the volume of calculations. The bipolar electrode with a diameter 10 mm were placed on the muscles according to the guideline of SENIAM [40]. The kinematics data were collected by Noraxon’s 3D myoMotion analysis system at the sampling rate of 100 Hz which was synchronized with myoMuscle system. It consists of a set of eight inertial sensors to capture lower extremity and pelvis 3D kinematics (myoMotion Clinical, Noraxon USA Inc.). Sensors were placed on foots, legs, thighs, and pelvis according to manufacture guide [39] (Figure 1).

### 2.2. Data Pre-Processing

First, a 20–400 Hz band-pass fourth-order Butterworth filter was used to remove any noise that occur above 400 Hz frequency and motion artifacts below 20 Hz from the sEMG signal. Most of the neural information exists in 50 Hz spectrum and using notch filter in that frequency will attenuate most of the information and not recommended. The kinematics data was filtered using fourth-order low-pass Butterworth filter with a cutoff frequency of 6 Hz. Then the kinematics and sEMG time series data in each loading condition were separated so that we had 20 cycles for ankle and knee joints in four different loading conditions for each participant. The signal segmentation was carried out using MATLAB built-in function “Islocalmin”. Through this function, the local minimum of the kinematics signal was determined and kinematic and sEMG signals were segmented based on the local minimum and each segment was considered as a cycle. Then each cycle interpolated using spline method to 1000 data point. Since the amplitude of sEMG and kinematics signal are different, it causes problems in the optimization process and convergence to the global minimum may not occur. Therefore, both sEMG and kinematics signal amplitude normalized to a number between one and zero using below Equation (1).
(1)$$X_{normalize\;}=\frac{X_{real\;range}-X_{min}}{X_{max}-X_{min}}$$
where $X_{real\;range},\;X_{normalize\;}$, $X_{min}$ and $X_{max}$ are the dataset in real range, dataset in normalize range, minimum and maximum of the dataset in real range, respectively.
(2)$$X_{real\;range}=X_{normalize}\times \left(X_{max}-X_{min}\right)+X_{min}$$

After training network using train dataset, output of the network and the target are returned to their real range using Equation (2) to calculate the performance of the network on test dataset. The complete pre-processing stage can be seen in Figure 2.

### 2.3. PCA Analysis

In order to ensure the effect of different loading on lower limb kinematics, we performed Principal Component Analysis (PCA) on the both knee and ankle joints kinematics data. For this purpose, all data related to 19 subjects were placed in a matrix with 76 rows and 5000 columns (five cycles). Each of the four rows of this matrix corresponds to a subject in four loading condition and each column corresponds to a time point. Due to the similarity of the results related to the knee and ankle joints, only the information and figures related to the knee joint are included here. Figure 3a shows that the range of motion decreases with increase in load. In the Figure 3b, the x-axis and the y-axis represent the first and second principal component, respectively. The results of the PCA showed that the targets that we want to predict have distinct boundaries in the reduced features space. Therefore, the network must be highly generalizable to predict such targets.

### 2.4. Feature Extraction

#### 2.4.1. Wavelet Decomposition

Feature extraction from sEMG signal is performed in three domains of time, frequency, and time-frequency. Discrete wavelet transform (DWT) is a common method for feature extraction in the time-frequency domain. sEMG is a non-stationary signal and previous research has shown that feature extraction from sEMG using DWT has a much better performance than other methods such as short-time Fourier Transform [41]. In this method, the signal is decomposed into two subgroups of coefficient according to its frequency spectrum, approximation, and details. In sEMG, the subgroup of details that has a high frequency spectrum behaves similar to noise and does not provide information. However, the coefficients related to approximation subgroup provide extensive information about the signal behavior and is the most important part [42]. As it has shown in Figure 4a, the signal decomposition using wavelet method can be carried out in several levels. Increasing the levels causes the high frequency components to be separated from the signal in the form of detail coefficients and the approximation components will have a smoother behavior. We performed DWT at different levels using the sym8 wavelet and the best result was related to the coefficients of the eighth levels approximation decomposition which is shown in Figure 4b.

#### 2.4.2. Cross-Correlation Analysis

The sEMG signal starts about 20–200 milliseconds earlier than the kinematic signal and this valuable feature gives us the ability to predict motion. However, this feature makes the two signals not completely in phase. Therefore, prediction of lower limb joint angles without considering the kinematic signal delay reduces the network accuracy and the desired output won’t achieved. In order to solve this problem, the cross-correlation method was used. In this method, the correlation between the two signals is measured for different delays. Cross-correlation is calculated by the following equations,
(3)$$\frac{1}{T}\;\sum_{t=1}^{T-k}\left(y_{1t}-{\bar{Y}}_{1}\right)\left(y_{2,\;t+k}-{\bar{Y}}_{2}\right),\;\;k=0,\;1,\;\ldots$$
(4)$$\frac{1}{T}\;\sum_{t=1}^{T+k}\left(y_{2t}-{\bar{Y}}_{2}\right)\left(y_{1,\;t-k}-{\bar{Y}}_{1}\right),\;\;k=0,\;-1,\;...$$
where C is the cross-covariance between two signals; $y_{1}$ and $y_{2}$, k is the lag considered. also ${\bar{Y}}_{1}$ and ${\bar{Y}}_{2}$ are the mean of two signal.
(5)$$Std_{y1}=\sqrt{C_{y1y1}^{\left(k=0\right)}}$$
(6)$$Std_{y2}=\sqrt{C_{y2y2}^{\left(k=0\right)}}$$
(7)CCy1y2(k)=Cy1y2(K)Stdy1Stdy2; k=0, ±1, …
where cross covariance of the same signal with zero lag is equal to Standard deviation (Std) of that signal and CC is the cross-correlation between two signals. Figure 5 shows the CC coefficient between sEMG signal of the vastus medialis muscle and the knee joint kinematics in different lag. In this figure, the dashed blue line indicates the degree of correlation at zero delay, and the solid blue line indicates the delay for which the correlation is maximized.

Since the lag values varied for each sEMG channel, different channels had to move forward by different amounts. Under these conditions, the lengths of the feature vectors were different from each other and the target vector. Therefore, the input signals were considered as finite support signals and were set to zero before and after them. Under these conditions, as the sEMG channels moved forward, all feature vectors and target vectors were still having the same length. Implement feature vector in finite support signal is shown in Figure 6.

#### 2.4.3. Kinematics Estimation

##### LSTM Network

LSTM cell consists of four main parts of forget gate, input gate, output gate, and cell state. In LSTM cell, the flow of information is controlled through gates. Cell state is a connection that enters information from the past into the cell. For preventing the vanishing or exploding of the gradient no mathematical operations are performed in the cell state.

Through the gates that are considered inside the LSTM cell, the importance of the past in the cell state is determined. The cell state is similar to the paths provided in the residual network (ResNet) for gradient flow [43] and it solves the two problems of classical RNNs which mentioned earlier. LSTM performance is based on the following Equations (8)–(12). In all of the following equations, σ ( ) represents the sigmoid activation function. $x_{t}$ is input sequence at time step *t*, $h_{t-1}$ is the hidden unit at the time step t−1 and ⊙ represents elementwise multiplication.
(8)$$f_{t}=\sigma \left(W_{f}\cdot \left[h_{t-1},x_{t}\right]+b_{f}\right)$$
(9)$$i_{t\;}=\sigma \left(W_{i}\cdot \left[h_{t-1},x_{t}\right]+b_{i}\right)$$
(10)$$o_{t}=\sigma \left(W_{o}\cdot \left[h_{t-1},x_{t}\right]+b_{o}\right)$$
where $f_{t}$, $i_{t}$ and $o_{t}$ are forget, input and output gate, respectively. The value of these gates is a number between 0 and 1, which is calculated by $W_{f}$, $W_{i}$ and $W_{o}$, $b_{f}$, $b_{i}$ and $b_{o}$ which are weights matrix for the forget, input and output gate, and bias considered for the forget, input and output gate, respectively.
(11)$$C_{t}=f_{t}⊙\;C_{t-1}+i_{t}⊙(tanh\left(W_{c}\cdot \left[h_{t-1},x_{t}\right]+b_{c}\right)$$
where $C_{t}$ is cell state at time step *t,* calculated by $f_{t}$, cell state at time step t−1 ($C_{t-1})$, $i_{t}$, $W_{c}$ and $b_{c}$ which $W_{c}$ and $b_{c}$ are the weights matrix and bias considered for cell state, respectively.
(12)$$h_{t}=o_{t}⊙\;tanh\left(C_{t}\right)$$

Finally, $h_{t}$ is the hidden unit at time step t which is cell output and calculated by elementwise multiplication between $o_{t}$ and *tanh*(⋯) of cell state at same time step.

The estimation was performed by a network consisting of 10 hidden layers. In each hidden layer 200 nodes were considered. For optimization and minimizing loss function, Adam′s algorithm was used due to its high speed in convergence compared to other algorithms with max epochs of 200 and learning rate of 0.005. Dropout method was used to prevent overfitting with drop probability of 0.2 in each hidden layer. Output layer of the network consists of two unit which are knee and ankle joint angle. In addition, the network input layer consists of 5 nodes. The number of rows and columns of the network input matrix is as follows.
(13)5 cycle×4 condition=20
(14)19 subject×20 cycle=380
(15)380×1000 data point=380,000

Based on the above, the network input was a matrix with 5 rows and 380,000 columns. It must be noted that for prediction independent of loading and activity intensity, as well as independent prediction of the subject which is main purpose of this study, only muscle sEMG signal should be considered as feature vector and the data related to all subjects in four loads must concatenated in a row. All the preprocessing stage and model implementation was carried out using MATLAB 2020b software (The MathWorks Inc., Natick, MA, USA). Neural network training process was conducted using NVIDIA GeForce^®^ RTX 2060 SUPER with 2176 CUDA core (NVIDIA CORPORATE, Santa Clara, CA, USA).

##### Multilayer Perceptron (MLP) Network

A three-layer MLP network was used to compare the results of LSTM network with classical networks. This network consists of one hidden layer with 160 nodes, an input layer with five nodes and an output layer with two nodes, which are related to the knee and ankle joint angle. In Levenberg–Marquardt algorithm, Jacobean training not supported on GPU therefore we set training function on TRAINSCG function that updates weight matrix according to the scaled conjugate gradient method. Hyperbolic tangent (tanh) function was considered as network activation function in hidden layer.

##### Model Selection and Network Accuracy Criteria

LSTM and MLP models with different hyperparameters were trained and tested by K-fold cross-validation method. After determining the optimal model, hyperparameters related to chosen model were considered and were reported. Then, Leave-one-out cross-validation method was used to evaluate the accuracy of the selected network. In this method, each time the model was trained by 18 subjects and then tested for subject 19. This process was repeated 19 times until the model was tested on all subjects and the outputs were concatenated together.

In this study, the root mean square error (RMSE) Equation (16) and correlation coefficient (r) (Equation (17)) were used to evaluate the prediction accuracy of both networks.
(16)$$RMSE=\sqrt{\frac{\sum_{i=1}^{N}{\left({\tilde{\theta }}_{i}-\theta_{i}\right)}^{2}}{N}}$$
(17)$$r=\frac{\frac{1}{N}\sum_{i=1}^{N}\left(\theta_{i}-\bar{\theta }\right)\left({\tilde{\theta }}_{i}-\bar{\tilde{\theta }}\right)}{\sqrt{\frac{1}{N}\sum_{i=1}^{N}{\left(\theta_{i}-\bar{\theta }\right)}^{2}}\sqrt{\frac{1}{N}\sum_{i=1}^{N}{\left({\tilde{\theta }}_{i}-\bar{\tilde{\theta }}\right)}^{2}}}$$
where ${\tilde{\theta }}_{i}$ is the model output, and $\theta_{i}$ is the target vector at the sampling time *i* and *N* is the length of output and target vectors.

## 3. Results

Figure 7 shows the prediction results of the LSTM network extracted from the Leave-one-out cross-validation method and the real kinematics of the knee (a) and ankle (b) joint for 19 subjects. In this figure, solid black line is the real joint angle and dashed red line is the LSTM output. The LSTM prediction showed in this figure was derived from the results of the 19 networks that were concatenated together in this diagram. Based on the results, the LSTM network accurately estimated the signals associated with each subject. Therefore, due to the difference in kinematic signal between different subjects and different conditions, the generalizability of the LSTM network is high. In the Figure 8, prediction results of MLP and LSTM network for knee and ankle joint for subject 1 in four loading conditions are shown. In this figure, solid black line is real joint angle, solid red line is LSTM output and solid blue line is MLP output. Due to the high number of subjects and lack of space, only the best and worst accuracy of network predictions for knee and ankle joints and the average results of 19 subjects have been reported in Table 1 and Table 2.

The accuracy of the LSTM and MLP networks for knee joint that evaluated by RMSE were 6.774 ± 1.197 and 9.489 ± 0.922, respectively (Table 1). The RMSE for ankle prediction by LSTM and MLP were 6.961 ± 1.200 and 9.705 ± 0.978, respectively (Table 1). Table 2 also shows the accuracy of both network based on *r* results. The accuracy of the LSTM and MLP networks for knee joint were 0.938 ± 0.0135 and 0.897 ± 0.013, respectively. The accuracy of the LSTM and MLP networks for ankle joint were 0.922 ± 0.012 and 0.882 ± 0.019, respectively.

In this study, we used the cross-correlation method to minimize the delay between the two signals and to maximize the correlation. This method is easy to use and its effect on network performance is shown in Figure 9. Using this method and based on *r* results, LSTM network with inputs that processed by CC method have a 3.8% and 4.7% better performance in knee and ankle joints, respectively, compared to the situation of LSTM in preprocessing stage that CC method was not used. In the MLP network, the results are similar and the performance of this network improved by 6.5% in the knee joint and 6% in the ankle joint using the CC method. Therefore, it can be concluded that using this method and considering delay between both signals, we can increase the accuracy of the prediction regardless of what network is used.

## 4. Discussion

The aim of the present study was to continuously predict the kinematics of the lower limb during real loading squat task that this prediction were considered in two conditions:Prediction regardless of the intensity of movement and loading conditionInter-subject prediction.

One of the main parameters to increase muscle strength and athletic performance is the increase of exercise intensity during different training sessions. This causes the body to adapt to the new intensity and improving the performance. Then prediction during different athletic intensities can help to design the prosthesis that works during different athletic situation. Moreover, inter-subject prediction has two major advantages, reducing the cost of purchasing and mass-producing prostheses. Therefore, using a myoelectric prosthesis in athletic activities can be used when condition (1) and (2) is met. Our results showed that the proposed network is able to meet both conditions and predict our target parameters with high accuracy under above considerations.

Our third purpose was to compare a deep network performance with a classical network in a regression task. In both joints, the prediction accuracy and generalizability of the deep network was higher than the classical counterpart. Our results showed that LSTM performance is better than MLP in both knee and ankle joints and this is in line with previous researches that compared the performance of deep and classical networks in their research [37,38]. However, the difference between the performances of the two networks was not what we expected. The reason for the slight superiority (average 4%) of LSTM over MLP can be considered the short length of our data set. Deep networks are algorithms that their performance is constantly increasing as the length of the data set increases. In biomechanical research, our data volume is acceptable, but from a data science perspective, the deep network trained in this research requires larger data set. Moreover, the prediction accuracy of both networks was better in the knee joint compared to the ankle joint (1.6% for LSTM and 1.5% for MLP as average). It should be noted that the inputs of both networks are related to five muscles. Four inputs act on the knee joint (MG, RF, VM and BF), but only two inputs act on the ankle joint (MG and TA). Therefore, the number of network inputs is the main factor in reducing the accuracy of prediction in the ankle joint. In addition, with increasing the number of muscles that work on ankle joint as the network inputs, the accuracy of prediction will increase in this joint.

The DWT method was used in the feature extraction part. The capability of this method in comparison with other feature extraction techniques has been investigated in previous studies and was not addressed in this paper. Our hypothesis derived from the theoretical issues related to the existence of a delay between the triggering neural message and joint kinematics was examined by the cross-correlation method. According to the results, considering the delay that maximize the correlation between two signals has a high effect in the accuracy of network. The cross-correlation method is simple to use and with the best of our knowledge it has not been considered in any previous research.

The prediction of kinematic parameters through biological signals in athletic tasks has not considered before and more of researches focused on daily activities such as gait or basic rehabilitation movements. Therefore, it is hard to compare our results with the findings of previous results. However, we tried to compare our results with the previous researches in Table 3. The comparison between our work and previous similar research is based on five features:The number of subjects and the volume of dataset.The number of muscles that are considered as network input.Proposed networks (In most reviewed paper, other networks have been used to compare the performance of the proposed network. Here, only the performance of the proposed network has been reported.).Target kinematics parameter.Network evaluation criteria and reported results.

According to our main goal -generalizability evaluation- we tried to use more participants in this study compared to previous research to reach more input network. The number of inputs of the proposed network is 5, which is equal to the number of inputs considered in the research of Xia et al. (2018) [36]. The number of muscles considered in the other two studies were 10 [28] and 7 [37], respectively. Reducing the number of inputs is an advantage because it reduces redundancy due to the overlap of muscle participation in a joint, which in turn increases the rank of the network input matrix. If muscles from a synergic group (for instance, extensors or flexors) are used, the redundancy elimination techniques are needed, which in turn increases the volume of calculations. The second advantage is the reduction in the number of sEMG sensors, which in turn leads to a reduction in manufacturing costs of the prostheses. In both Xia et al. (2018) and Chen et al. (2019) studies [36,37], researchers have used deep networks, but their target parameter has been the upper limb. But in the Chen et al. (2018) [28] study, the lower limbs were considered and only classical networks were used. In line with our results, in the study conducted by Chen et al. (2018) [28], network performance for the ankle joint decreased by about 1% compared to the knee joint. The results of this study were 3.2% better for the knee joint and 3.8% for the ankle joint compared to the results of our study. In two other studies, the upper limb motion has been considered. The degrees of freedom in the upper limb are higher, which makes network performance more challenging. However, the lower limbs support the weight of the whole body and are in constant contact with the earth and external forces that implies the importance of precise kinematic adjustments in lower limb prostheses. The performance of proposed network for hand position prediction [36] was 90.3%, which is 3.5% weaker compared to the result of the present study. Moreover, the reported result by Chen et al. (2019) was 8.3% weaker compared to our average result in the knee joint [37]. In study conducted by Ma et al. (2020) the LSTM network has been used to estimate knee joints from sEMG of eight lower extremity muscles in five participants during treadmill gait. According to the coefficient of determination ($R^{2}$) index, the best reported result in this research was 98.44% [44]. Compared to the results of Ma et al. (2020), the results reported for the knee joint in the present study is about 4% weaker. However, in present study the squat task (a real athletic condition) was considered under four different loading conditions that may lead to reduced network performance.

Finally, we had some limitations regarding this research. As mentioned above, our dataset length is suitable for biomechanical research and classical networks. Nevertheless, increasing the data volume for deep networks will improve network performance and will definitely increase the gap between deep and classic networks. Another limitation of this research was the lack of prediction of kinetic parameters and joint torques, which were not possible due to not using the force plate and inverse dynamics calculations. In addition to kinematic adjustments, the prediction of joint torques in the control of a myoelectric prosthesis should be of interest to researchers.

## 5. Conclusions

In this study, a deep recurrent neural network was used to predict the kinematics of the lower limb during squat training in real loading conditions with the aim of controlling myoelectric prostheses in athletic movements. The results showed that the proposed network is able to predict lower limb kinematics in four different loads with high accuracy (based on Correlation Coefficient index 92.2% and 93.8% for ankle and knee joint, respectively). In addition, with the aim of minimizing the delay between sEMG and kinematic signals and increasing the accuracy of the network, the cross-correlation method was used. Considering the delay between both signals and using this method in the data preprocessing stage increased the performance of the proposed network by 3.8% and 4.7% for knee and ankle joint, respectively. The results of present study can be used in the myoelectric prostheses when performing athletic movements such as squat.

## Figures and Tables

**Figure 1 sensors-21-07773-f001:**
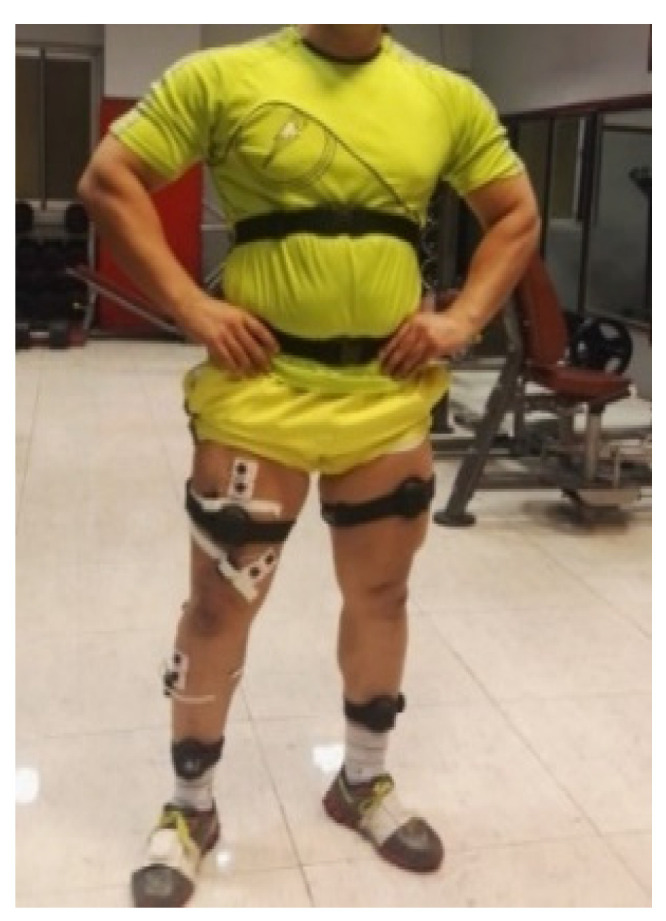
Noraxon’s myoMotion and myoMuscle sensor location.

**Figure 2 sensors-21-07773-f002:**
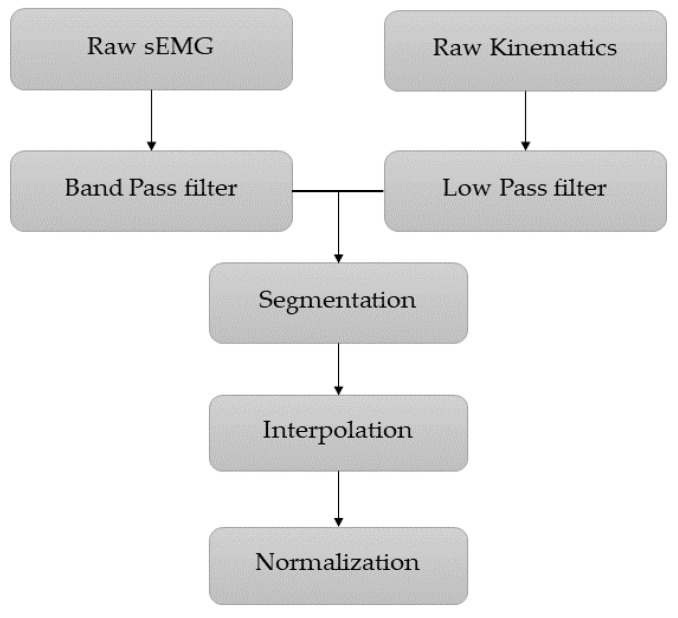
Block diagram of the pre-processing stage.

**Figure 3 sensors-21-07773-f003:**
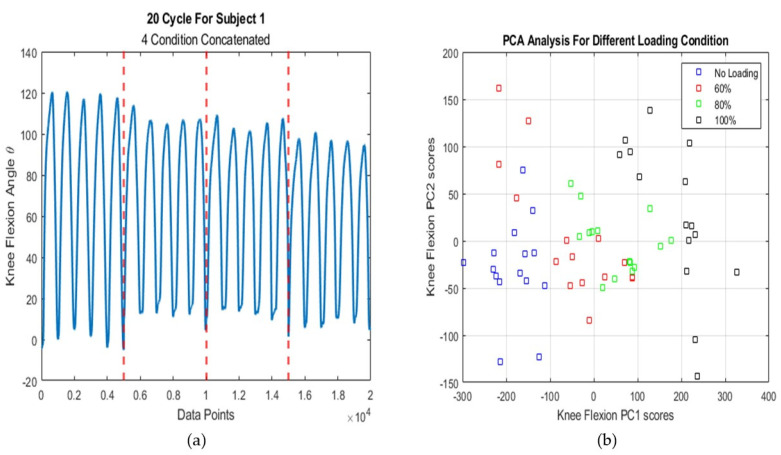
(**a**) Effect of Loading on Kinematics Signal in Knee Joint Flexion; (**b**) PC scores for Different Loading Condition for Knee Joint.

**Figure 4 sensors-21-07773-f004:**
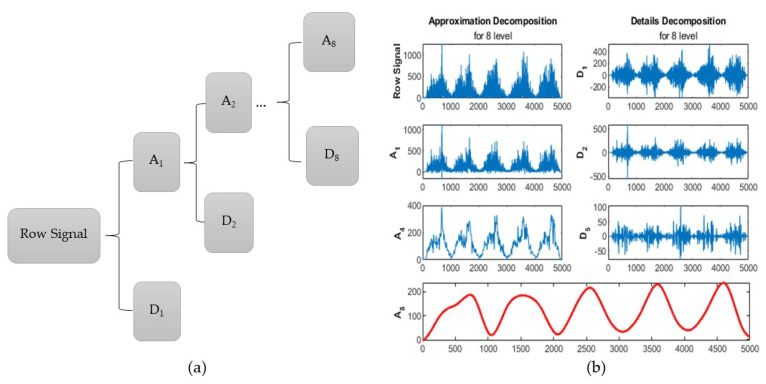
Discrete wavelet transform (DWT) (**a**) decomposition tree; (**b**) decomposition of sEMG signal using wavelet technique.

**Figure 5 sensors-21-07773-f005:**
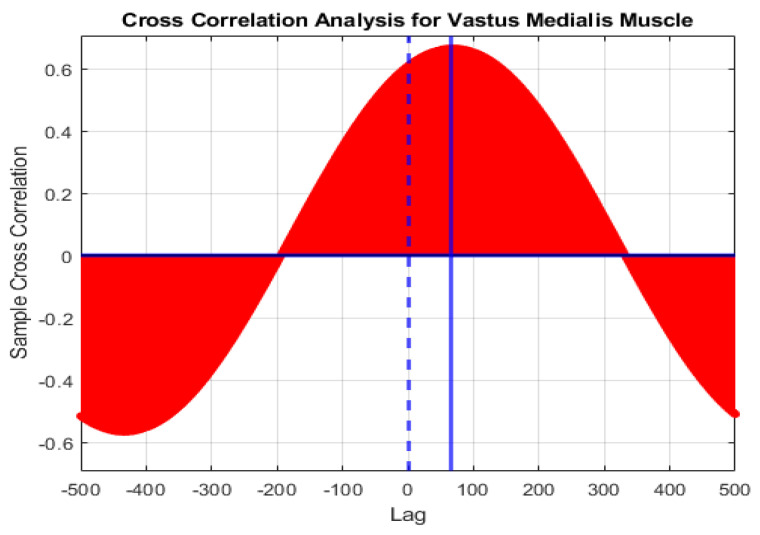
Cross-correlation (CC) of the sEMG signal of vastus medialis muscle.

**Figure 6 sensors-21-07773-f006:**
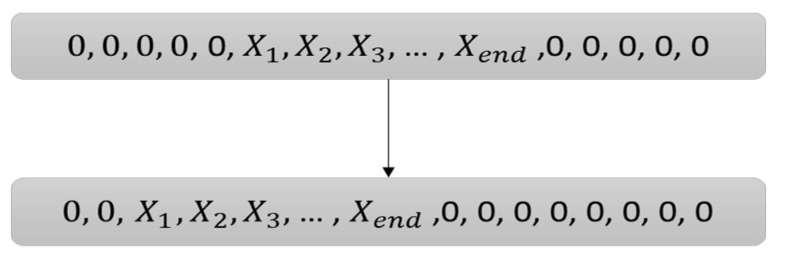
Lag consideration in feature vector in finite support signal.

**Figure 7 sensors-21-07773-f007:**
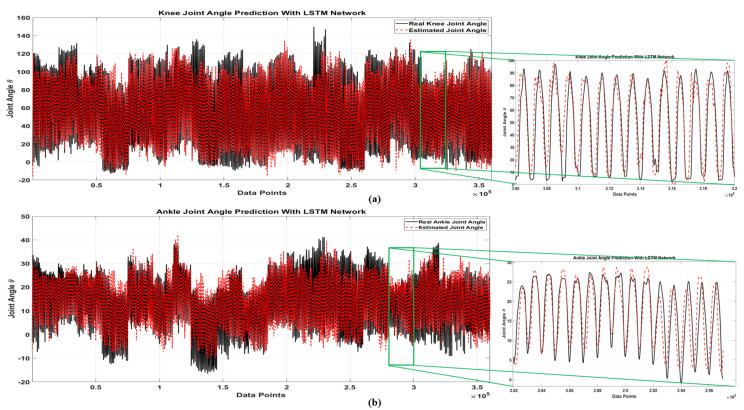
Full Data Prediction with LSTM Network; (**a**) Knee Joint (**b**) Ankle Joint.

**Figure 8 sensors-21-07773-f008:**
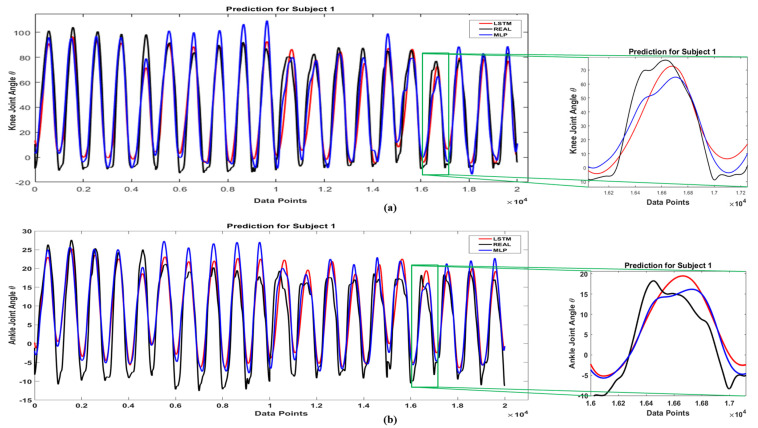
Full Data Prediction with LSTM Network; (**a**) Knee Joint (**b**) Ankle Joint.

**Figure 9 sensors-21-07773-f009:**
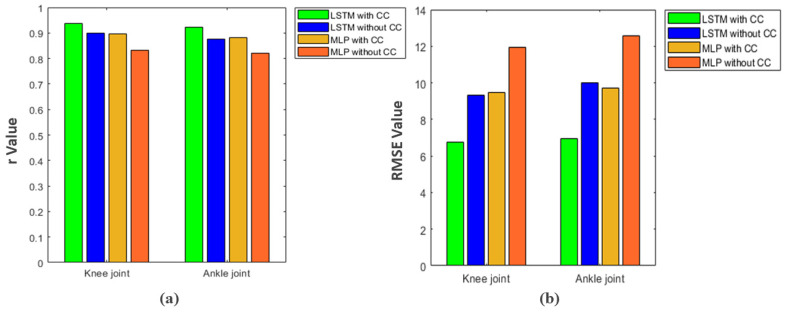
Investigating the effect of cross-correlation analysis and considering the delay of sEMG signal in two networks in both joints, (**a**) correlation coefficient value (*r*); (**b**) RMSE value.

**Table 1 sensors-21-07773-t001:** Root mean squared error (RMSE) results for both model in two joints.

Subject	LSTM		MLP	
	Knee Joint	Ankle Joint	Knee Joint	Ankle Joint
Best	5.537	5.716	8.577	8.721
Worse	7.932	8.120	10.42	10.70
Average	6.774 ± 1.197	6.961 ± 1.200	9.489 ± 0.922	9.705 ± 0.978

**Table 2 sensors-21-07773-t002:** Correlation coefficient (*r*) results for both model in two joints.

Subject	LSTM		MLP	
	Knee Joint	Ankle Joint	Knee Joint	Ankle Joint
Best Result	0.954	0.941	0.910	0.901
Worst Result	0.927	0.916	0.879	0.865
Average	0.938 ± 0.0135	0.922 ± 0.012	0.897 ± 0.013	0.882 ± 0.019

**Table 3 sensors-21-07773-t003:** Previous studies and their findings.

	Number of Subjects	Proposed Model	Input of the Model	Target Parameters	Accuracy Criteria and Performance
Present Study	19	(1) LSTM	VM, RF, BF, TA and MG	(a) Ankle (*θ*) (b) Knee (*θ*)	1-(a) RMSE: 6.961, *r*: 0.922 (92.2%) 1-(b) RMSE: 6.774, *r*: 0.938 (93.8%)
Xia et al. (2018) [36]	8	(1) CNN (2) RCNN	BB, TB, AD, PD and MD	Hand position in 3D	(1) *R*^2^: 77.6% (2) *R*^2^: 90.3%
Chen et al. (2018) [28]	6	(1) BP	RF, VL, VM, SR, AT, ST, BF, MG, VG and SL	(a) Ankle (*θ*) (b) Knee (*θ*) (c) Hip (*θ*)	1-(a) RMSE: 2.45, *r*: 0.96 1-(b) RMSE: 3.96, *r*: 0.97 1-(c) RMSE: 3.58, *r*: 0.95
Chen et al. (2019) [37]	7	(1) LSTM	BR, BB, TB, PD, MD, AD and PM	Shoulder	(1) RMSE: 6.1833 (1) *R*^2^: 0.8556
Ma et al. (2020) [44]	5	(1) LSTM	RF, BF, ST, GC, SM, SR, MG, TA	Knee (*θ*)	(1) RMSE: 3.472 (1) *ρ*: 98.44

LSTM: Long Short Term Memory, RCNN: Recurrent Convolutional Neural Network, CNN: Convolutional Neural Network, BP: Back Propagation; RF: Rectus Femoris, BF: Biceps Femoris, ST: Semitendinosus, GC: Gracilis, SM: Semimembranosus, SR: Sartorius, MG: Medial Gastrocnemius, TA: Tibialis Anterior, VM: Vastus Medialis, VL: Vastus Lateralis, SL: Soleus, LG: Lateral Gastrocnemius, BB: Biceps Brachii, TB: Triceps Brachii, AD: Anterior Deltoid, PD: Posterior Deltoid, MD: Medial Deltoid, BR: Brachioradialis, BB: Biceps Brachialis, TB: triceps Brachialis, PM: Pectoralis Major; RMSE: Root-mean-square Error, r: Correlation Coefficient, *R*^2^: the coefficient of determination.

## Data Availability

Data is not available for this research.

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
