# Peer review of "Estimation of Lower Limb Kinematics during Squat Task in Different Loading Using sEMG Activity and Deep Recurrent Neural Networks"

_sensors, 2021, doi:10.3390/s21237773_

Round 1

Reviewer 1 Report

Congrats for the study conception, it is very interesting and useful.

However, there are some points that can be improved.

Introduction

Quite elucidative and clear, the gap is evidenced.

P2 L46

Small caps – “winter and sienko (1988)”

P3L145

Why did you choose body builders? According to your answer, I will suggest adding this complement to the methods.

P4 L161

In my opinion, figure 1 could bring more details of the protocol beyond that are shown, e.g., the body builder performing a squat movement to better elucidate the movement tested.

P5 L192

A small detail in figure 2, is missing a horizontal line.

P14 L428

“(i) Prediction regardless of the intensity of movement and the occurrence of fatigue”

I do not see any mention of muscle fatigue protocol, it is a fragile point that must be fixed. But, considering the protocol with 2 min of rest, how can you be sure that the muscles were fatigued? Commonly, we show muscle fatigue by muscle force decrease in MVC tests, no matter if they are isometric or dynamic.

P15 L434

From here, it is important to fit the idea considering your findings and be sure that you can talk about muscle fatigue.

P15 L472

“According to our main goal -generalizability evaluation- the number of subjects considered in this study is more than all previous researches.”

If you consider talking about this thing here in the discussion, better than this, is to show the calculation of the sample size in the methods section.

Author Response

Response to Reviewer Comments

(Manuscript ID: sensors-1428435)

We would like to thank the reviewers for their constructive reviews and we appreciate their comments on our manuscript. We have addressed all comments below and have revised our manuscript according to the reviewer’s comments. We have highlighted changes to our manuscript in red font.

Reviewer1

Congrats for the study conception, it is very interesting and useful.

However, there are some points that can be improved.

Comment 1: Introduction: Quite elucidative and clear, the gap is evidenced.

Response: Thank you for your positive comment.

Comment 2: P2 L46: Small caps – “winter and sienko (1988)”

Response: It is edited in the reviewed manuscript.

Comment 3: P3 L145: Why did you choose body builders? According to your answer, I will suggest adding this complement to the methods.

Response: Our main aim was to predict joints kinematics from sEMG in real loading situation and we chose trained body builder in this study because they were familiar with squat task in real different loading situation and they should press high loads in their testing. This explanation was added in the reviewed manuscript to address the reviewer comment.

Comment 4: P4 L161: In my opinion, figure 1 could bring more details of the protocol beyond that are shown, e.g., the body builder performing a squat movement to better elucidate the movement tested.

Response: Thank you for your precise comment. Unfortunately, we did not have photo of participants with sEMG and MyoMotion sensors during squat task to add in this manuscript.

Comment 5: P5 L192: A small detail in figure 2, is missing a horizontal line.

Response: Thank you for your precise comment. It edited in the revised manuscript to address the reviewer comment.

Comment 6: P14 L428: “(i) Prediction regardless of the intensity of movement and the occurrence of fatigue”

I do not see any mention of muscle fatigue protocol, it is a fragile point that must be fixed. But, considering the protocol with 2 min of rest, how can you be sure that the muscles were fatigued? Commonly, we show muscle fatigue by muscle force decrease in MVC tests, no matter if they are isometric or dynamic.

Response: Thank you for your precise comment. We accept the reviewer comment and as we did not measure fatigue, this sentence was deleted in the reviewed manuscript.

Comment 7: P15 L434: From here, it is important to fit the idea considering your findings and be sure that you can talk about muscle fatigue.

Response: Thank you for your precise comment. We accept the reviewer comment about “fatigue”. This sentence was edited in the reviewed manuscript.

Comment 8: P15 L472: “According to our main goal -generalizability evaluation- the number of subjects considered in this study is more than all previous researches.”

If you consider talking about this thing here in the discussion, better than this, is to show the calculation of the sample size in the methods section.

Response: We did not use any sample size calculation method in our study. We just test the professional bodybuilder who voluntarily accepted to participate in our study. This sentence was edited in the reviewed manuscript to address the reviewer comment.

Reviewer 2 Report

Thank you for the opportunity to review the manuscript entitled “Estimation of Lower Limb Kinematics Based on sEMG Activity Using Deep Recurrent Neural Networks During Squat Task in Different Loading”. This topic is very interesting and it’s well-organized study. Below please find out specific comments and suggestions.

Specific comments

Page 1, line 14 and line 20: “sEMG”, “CC method”

  • The full spelling of sEMG and CC method should be provided with the first appearance of the abbreviation.

Page 2, paragraph 2, line 56-58: “The sEMG is rich of neural information ~ however, it is stronger than EEG signal and it is a non-invasive method.”

  • In the case of hand prosthesis, studies on motion control or classification using EEG have been performed. Shall we conclude that sEMG is a stronger signal to control motion than an EEG signal?

Page 2, paragraph 2, line 62-64: “Previous research has focused on ~ stair ascending and descending”

  • Please provide a reference.

Page 3, paragraph 2, line 113-114: “The problem of exploding or vanishing gradients with increasing layers in the back propagation process and the problem of short-term memory.”

  • Please consider rewriting the sentence.

Page 3, paragraph 1, line 145 – Page 4, paragraph 1, line 163: “Nineteen trained body builders with no history of lower extremity injuries ~ placed on foots, legs, thighs, and pelvis according to manufacture guide (Figure 1)”

  • Please provide content about IRB approval.

Page 4, paragraph 1, line 154-158: “These muscles are most involved in knee and ankle joint movements ~ that increase computational volume.”

  • Please add a reference.

Page 4, paragraph 1, line 158-159: “The bipolar electrode with a diameter 10 mm were placed on the muscles according to the guideline of SENIAM.”

  • Please provide a reference.

Page 4, paragraph 2 line 175-176: The cycle separation was done using MATLAB function islocalmin with min prominence of 10.”

  • What does islocalmin mean? Please use precise terminology and describe this method clearly.

Page 5, paragraph 1, line 197-205: In order to ensure the effect of different loading ~ So, the network must be highly generalizable to predict such targets.

  • The author reported that the effect of different loading on lower limb kinematics based on their similarity using Principal Component Analysis. In this text, however, only the separation results according to the knee joint and loading were reported. Regarding the ankle joint, what is the separation result according to different loading?

Page 6, paragraph, line 228-230: “We performed DWT at different levels using the sym8 wavelet ~ eighth levels approximation decomposition which is shown in Figure 3b.”

  • What does sym8 mean? Is it one of the wavelet types?

Results

Page 13, paragraph 1, line 398-399: “The RMSE for ankle prediction by LSTM and MLP were 0.922±0.012 and 9.705±0.978, respectively (table 1).”

  • The RMSE value reported in the main text and Table 1 do not match.

Page 13, paragraph 1, line 403 ~ Page 14, page 14, paragraph 1, line 409: “It should be noted that the inputs of both networks ~ We expected a performance increase of about 10%, which was not seen in results of this study.”

  • These contents do not fit into the Results section. Please consider moving these contents to the Discussion section.

Page 14, paragraph 2, line 410-417: “In this study, we used the cross-correlation method ~ we can increase the accuracy of the prediction regardless of what network is used.”

  • There is an error in the resulting graph of Figure 10. It seems that the result graphs of RMSE and r are switched. Please correct the graph.
  • Please only report results in the Results section. For the rest, move it to the discussion section.

Page 15, paragraph 4, line 459 – Page 16, paragraph 1, line 496: “The prediction of kinematic parameters through biological signals in athletic tasks ~ compared to our average result in the knee joint [38].”

  • The author mentioned that the degrees of freedom in the upper limb are higher, which makes network performance more challenging, and the lower limbs support the weight of the whole body and are in constant contact with the earth and external forces that implies the importance of precise kinematic adjustments in lower limb prostheses. From these things, it seems appropriate to compare this study with the study predicting the kinematics of the lower limb rather than the study predicting the kinematics of the upper limb.

  • No mention of the sequence labeling task. Did you predict joint kinematics with unsupervised learning?

  • I suggest you add conclusions about this study.

Author Response

Response to Reviewer Comments

(Manuscript ID: sensors-1428435)

We would like to thank the reviewers for their constructive reviews and we appreciate their comments on our manuscript. We have addressed all comments below and have revised our manuscript according to the reviewer’s comments. We have highlighted changes to our manuscript in red font.

Reviewer 2

Thank you for the opportunity to review the manuscript entitled “Estimation of Lower Limb Kinematics Based on sEMG Activity Using Deep Recurrent Neural Networks During Squat Task in Different Loading”. This topic is very interesting and it’s well-organized study. Below please find out specific comments and suggestions.

Specific comments

Comment1: Page 1, line 14 and line 20: “sEMG”, “CC method”

The full spelling of sEMG and CC method should be provided with the first appearance of the abbreviation.

Response: This information were edited to address the reviewer comment.

Comment 2: Page 2, paragraph 2, line 56-58: “The sEMG is rich of neural information ~ however, it is stronger than EEG signal and it is a non-invasive method.”

In the case of hand prosthesis, studies on motion control or classification using EEG have been performed. Shall we conclude that sEMG is a stronger signal to control motion than an EEG signal?

Response: The authors thank the reviewer for this precise comment. This sentence was edited to address the reviewer comment.

Comment 3: Page 2, paragraph 2, line 62-64: “Previous research has focused on ~ stair ascending and descending”

Please provide a reference.

Response: We provided a reference to address the reviewer comment.

Comment 4: Page 3, paragraph 2, line 113-114: “The problem of exploding or vanishing gradients with increasing layers in the back propagation process and the problem of short-term memory.”

Please consider rewriting the sentence.

Response: This sentence was rewritten to address the reviewer comment.

Comment 5: Page 3, paragraph 1, line 145 – Page 4, paragraph 1, line 163: “Nineteen trained body builders with no history of lower extremity injuries ~ placed on foots, legs, thighs, and pelvis according to manufacture guide (Figure 1)”

Please provide content about IRB approval.

Response: The authors thank to this precise comment. The IRB approval was provided in the text to address the reviewer comment.

Comment 6: Page 4, paragraph 1, line 154-158: “These muscles are most involved in knee and ankle joint movements ~ that increase computational volume.” Please add a reference.

Response: We provided a reference to address the reviewer comment.

Comment 7: Page 4, paragraph 1, line 158-159: “The bipolar electrode with a diameter 10 mm were placed on the muscles according to the guideline of SENIAM.”

Please provide a reference.

Response: This sentence was provided with a reference to address the reviewer comment.

Comment 8: Page 4, paragraph 2 line 175-176: The cycle separation was done using MATLAB function islocalmin with min prominence of 10.”

What does islocalmin mean? Please use precise terminology and describe this method clearly.

Response: This terminology and the method described in the reviewed manuscript to address the reviewer comment.

Comment 9: Page 5, paragraph 1, line 197-205: In order to ensure the effect of different loading ~ So, the network must be highly generalizable to predict such targets.

The author reported that the effect of different loading on lower limb kinematics based on their similarity using Principal Component Analysis. In this text, however, only the separation results according to the knee joint and loading were reported. Regarding the ankle joint, what is the separation result according to different loading?

Response: We run PCA on knee and ankle joints data and the results were similar. Then the results of ankle joint was not included in the article because of similarity and increasing the number of figures. This result is presented in the below figure and if it is necessary, we can add this figure in the manuscript.

Figure: PC scores for Different Loading Condition for Knee Joint

Comment 10: Page 6, paragraph, line 228-230: “We performed DWT at different levels using the sym8 wavelet ~ eighth levels approximation decomposition which is shown in Figure 3b.”

What does sym8 mean? Is it one of the wavelet types?

Response: There are different families of wavelets, such as haar, daubechies, symlets etc. The symlets are nearly symmetrical wavelets proposed by Daubechies as modifications to the db family. The symlets 8 are one of the subgroup related to the symlets group. This wavelet in matlab called sym8.

Results

Comment 11: Page 13, paragraph 1, line 398-399: “The RMSE for ankle prediction by LSTM and MLP were 0.922±0.012 and 9.705±0.978, respectively (table 1).”

The RMSE value reported in the main text and Table 1 do not match.

Response: Thank you for your precise comment. This mistake report was edited in the reviewed manuscript.

Comment 12: Page 13, paragraph 1, line 403 ~ Page 14, page 14, paragraph 1, line 409: “It should be noted that the inputs of both networks ~ We expected a performance increase of about 10%, which was not seen in results of this study.”

These contents do not fit into the Results section. Please consider moving these contents to the Discussion section.

Response: This content moved to the discussion section to address the reviewer comment.

Comment 13: Page 14, paragraph 2, line 410-417: “In this study, we used the cross-correlation method ~ we can increase the accuracy of the prediction regardless of what network is used.”

There is an error in the resulting graph of Figure 10. It seems that the result graphs of RMSE and r are switched. Please correct the graph.

Response: Thank you for your precise comment. It is edited in the reviewed manuscript.

Please only report results in the Results section. For the rest, move it to the discussion section.

Response: This content moved to the discussion section to address the reviewer comment.

Comment 14: Page 15, paragraph 4, line 459 – Page 16, paragraph 1, line 496: “The prediction of kinematic parameters through biological signals in athletic tasks ~ compared to our average result in the knee joint [38].”

The author mentioned that the degrees of freedom in the upper limb are higher, which makes network performance more challenging, and the lower limbs support the weight of the whole body and are in constant contact with the earth and external forces that implies the importance of precise kinematic adjustments in lower limb prostheses. From these things, it seems appropriate to compare this study with the study predicting the kinematics of the lower limb rather than the study predicting the kinematics of the upper limb.

Comment 15: No mention of the sequence labeling task. Did you predict joint kinematics with unsupervised learning?

Response: Supervised algorithms have been used in this study. In addition, one network is used to predict the target signal at all loads instead of using classification algorithms to classify each loadings and use for estimation in each loading. In this way, it is possible to have a network that predicts the target parameter with high accuracy, regardless of the intensity of activity.

Comment 16: I suggest you add conclusions about this study.

Response: We added a conclusion in the reviewed manuscript to address the reviewer comment.

Round 2

Reviewer 2 Report

Thank you for the opportunity to review the revised version of the manuscript entitled “Estimation of Lower Limb Kinematics Based on sEMG Activity Using Deep Recurrent Neural Networks During Squat Task in Different Loading”. Overall, the revised version has adequately addressed the major points raised in the last review. However, the answer to comment 14 is missing, thus a re-answer is required.

Author Response

Comment 14: Page 15, paragraph 4, line 459 – Page 16, paragraph 1, line 496: “The prediction of kinematic parameters through biological signals in athletic tasks ~ compared to our average result in the knee joint [38].”

The author mentioned that the degrees of freedom in the upper limb are higher, which makes network performance more challenging, and the lower limbs support the weight of the whole body and are in constant contact with the earth and external forces that implies the importance of precise kinematic adjustments in lower limb prostheses. From these things, it seems appropriate to compare this study with the study predicting the kinematics of the lower limb rather than the study predicting the kinematics of the upper limb.

Response: This part was edited and was highlighted with red color in the reviewed manuscript to address the reviewer's comment. Nevertheless, I forgot to add this in the response to the reviewer's comment in the former "response to the reviewer comment". 
